# Clinical Value of Circulating miRNA in Diagnosis, Prognosis, Screening and Monitoring Therapy of Pancreatic Ductal Adenocarcinoma—A Review of the Literature

**DOI:** 10.3390/ijms24065113

**Published:** 2023-03-07

**Authors:** Jakub Wnuk, Joanna Katarzyna Strzelczyk, Iwona Gisterek

**Affiliations:** 1Department of Oncology and Radiotherapy, Faculty of Medical Sciences in Zabrze, Medical University of Silesia in Katowice, 35 Ceglana St., 40-515 Katowice, Poland; 2Department of Medical and Molecular Biology, Faculty of Medical Sciences in Zabrze, Medical University of Silesia in Katowice, 19 Jordana St., 41-808 Zabrze, Poland

**Keywords:** circulating miRNA, pancreatic cancer, diagnosis, screening, prognosis, therapy monitoring, pancreatic ductal adenocarcinoma

## Abstract

Pancreatic cancer (PC) is considered to be the seventh most common cause of cancer-related deaths. The number of deaths caused by PC is estimated to increase in the future. An early diagnosis of PC is crucial for improving treatment outcomes. The most common histopathological subtype of PC is pancreatic ductal adenocarcinoma (PDAC). MicroRNAs (miRNAs)—which are endogenous non-coding RNAs involved in the posttranscriptional regulation of multiple gene expression—constitute useful diagnostic and prognostic biomarkers in various neoplasms, including PDAC. Circulating miRNAs detected in a patient’s serum or plasma are drawing more and more attention. Hence, this review aims at evaluating the clinical value of circulating miRNA in the screening, diagnosis, prognosis and monitoring of pancreatic ductal adenocarcinoma therapy.

## 1. Introduction

Pancreatic cancer (PC)remains a leading cause of cancer-related deaths. It is estimated that PC was responsible for 466,003 deaths in 2020,which puts it as the seventhleading cause of cancer-related deaths [1]. The incidence of PC has been either stable or slightly increased, with the highest incidence in North America, Europe and Australia/New Zealand [2]. The 5year relative survival rate is 11.5%, and it is largely due to late diagnosis and asymptomatic course at early stages [3,4].The number of deaths caused by PC is predicted to increase to 111,500 by 2025 in the European Union. In that case, PC will become the third leading cause of cancer-related deaths in the EU [5].

The risk factors of PC are tobacco smoking, obesity, diabetes mellitus, a diet rich in processed meat and lack of physical activity. Moreover, several gene mutations contribute to an increased risk of pancreatic cancer (*p53*, K-ras, *p16*, PRSS1) [6,7].

The most common histopathological subtype of PC is pancreatic ductal adenocarcinoma (PDAC). It is estimated that it accounts for more than 90% of PC diagnoses. The other types of PC are considered to have an impact on prognosis but have a lesser impact on management decisions [8].

Early diagnosis of PC is crucial for improving treatment outcomes. Ultrasound (US), computed tomography (CT), magnetic resonance imaging (MRI) and endoscopic ultrasonography (EUS) with EUS-guided fine-needle aspiration (EUS-FNA) are conventional methods used in PC diagnosis [9]. The advantages and disadvantages of these diagnostic methods are presented in Appendix A. Novel biomarkers related to PC, such as circulating tumor extracellular vesicles, circulating microRNAs (miRNAs), circulating tumor cells (CTCs), cell-free circulating tumor DNAs (cfDNAs) and long noncoding RNAs (lncRNAs), emerged in recent years as markers that might be useful as diagnostics and prognostic factors [10,11].

MiRNAs are endogenous non-coding RNAs that are involved in posttranscriptional protein synthesis regulation and play a significant role in cancer initiation, progression, proliferation, metastasis and chemotherapy resistance. They bind to 3’ untranslated regions (3’ UTRs) of targeted transcripts causing transcript repression or degradation- through the formation of RNA-induced silencing complexes (RISCs) [12,13].The diagnostic and prognostic potential of different miRNAs has been proved in various neoplasms, such as cervical cancer, lung cancer, gastric cancer or colorectal cancer [14,15,16,17]. Apart from tissue samples, miRNA may be detected in bodily fluids, including saliva (miR-3679-5p, miR-940, miR-21, miR-23a, miR-23b and miR-29, for example) [18,19], pancreatic fluid [20], peritoneal lavage fluid [21], cerebrospinal fluid [22] or urine [23]. Furthermore, numerous studies provide evidence that blood-based miRNAs may play a role in PC diagnosis, prognosis and surveillance during treatment [24,25]. Therefore, circulating miRNAs have gained more attention in recent years, as minimally invasive cancer markers and an alternative and supplementary approach to more invasive diagnostic methods currently available.

## 2. Literature Search

The literature search and study selection followed PRISMA guidelines [26]. The articles dating from 2003 to 2022 that were included in this review were evaluated based on their titles and abstracts. The eligibility criteria were based on the PICOS framework: (1) Participants: patients with PDAC; (2) Interventions: the detection of circulating miRNAs (3) Comparisons: non-PC controls; (4) Outcomes: area under the curve (AUC) of receiver operating characteristic (ROC) analysis, diagnostic sensitivity (SEN) and specificity (SPE),prognostic value, screening value or surveillance value of miRNA and (5) Study design: diagnostic research.

A combination of entry terms, including “plasma”, “circulating”, “blood-based”, “free”, “exosomal”, “serum”, “miRNA”, “microRNA” and ”pancreatic cancer” was used to search the PubMed and EMBASE databases. We also performed a manual search of potentially eligible studies based on reference lists of extracted reviews. The search was performed from 5 to 16 September 2022.

The extracted data included: specimen type, the investigated miRNAs and their expression levels in PDAC patients, whether miRNAs were used as single markers or combined as the panel, normalization control, AUC of ROC in diagnostic utility analysis, diagnostic SEN and SPE of the test, test comparison to carbohydrate antigen 19-9 (CA19-9) SEN and SPE, usefulness of miRNA as a prognostic factor, screening performance of miRNA and disease surveillance with miRNA.

The flow diagram of the literature search and study selection process is presented in Figure 1.

The aim of this study was to provide a literature review on the diagnostic, prognostic, predictive, screening and monitoring of the therapeutic value of circulating miRNA in patients diagnosed with PDAC.

The risk of bias was assessed with the QUADAS-2 tool, which assesses the risk in four domains: “patient selection”, “index test”, “reference standard” and “flow and timing” [27]. We have recognized that studies evaluated in this review exhibited a high risk of bias in the “patient selection” domain since they included patients with a definitive diagnosis. The risk of bias was intermediate in the “index test” domain because articles included in this review were trying to establish the threshold for the studied miRNAs. The risk of bias in “reference standard” and “flow and timing” was considered low.

## 3. Diagnostic Performance of Circulating miRNA

There were 44 studies included in this review that covered the topic of the diagnostic value of 110 single circulating miRNAs [28,29,30,31,32,33,34,35,36,37,38,39,40,41,42,43,44,45,46,47,48,49,50,51,52,53,54,55,56,57,58,59,60,61,62,63,64,65,66,67,68,69,70]. In those studies, miRNA expression was evaluated with real-time quantitative polymerase chain reaction (RT-Q-PCR). Details of the studies and their results, with predicted targeted genes, based on “miRBase: the microRNA database” are included in Appendix A. A hundred and ten different single miRNAs were tested 145 times in included the 44 studies. The diagnostic value determined with receiver operating characteristic curves (ROC) and their areas under the curve (AUC) varied within those studies from 0.618 to 1.0. In 17.93% (*n* = 26) of cases the AUC value reached or exceeded 0.9. Sensitivity (SEN) and specificity (SPE), if provided in the selected studies, ranged from 58% to 100% and 50% to 100%, respectively. The samples for tests were collected from blood plasma in 53.34% of cases and from blood serum in 49.66%. In 15.86% of cases, the tested miRNA was down-regulated compared to controls and, in 84.14% of cases, up-regulated compared to controls. In 42.76% of cases (*n* = 62), an endogenous control was made to normalize the data; in 51.03% of cases, an exogenous control was employed (*n* = 4). Both methods were used in 2.07% of cases (*n* = 3) In 4.14% of cases (*n* = 6), we did not find precise information about normalization methods (S2).

In 25 studies [28,30,31,36,38,39,43,44,45,49,50,51,54,55,57,58,62,64,65,67,70,71,72,73,74], miRNA was combined with other miRNAs, protein markers or CA19-9 into 33 diagnostic panels. The AUC value in ROC analyses was increased and ranged from 0.7207 to 1.0 (with 2 of them lacking data on AUC). SEN and SPE varied from 64% to 100% and 73% to 100%, respectively. In 51.52% (n = 17), cases the AUC value reached or exceeded 0.9. For details, see Appendix A.

## 4. Diagnostic Performance of Circulating Single miRNA Compared to CA19-9

We reviewed 16 studies that involved an analysis of the diagnostic performance of single miRNA compared to the diagnostic utility of CA19-9 [28,29,34,36,44,48,53,54,55,58,63,64,66,68,69,74]. In those studies,55 single circulating miRNAs were evaluated 59 times with 66.10% of them presenting better diagnostic value than CA19-9 alone and 23.73% worse than CA19-9 alone. In 6 cases there were no data to evaluate the diagnostic value compared to CA19-9 alone. In all cases with a worse diagnostic value of miRNA compared to CA19-9, an analysis of panels combining tested miRNA and CA19-9 was conducted. It proved that this combination improves the diagnostic performance of miRNA. Details are presented in Appendix A.

Three of the analyzed studies evaluated the expression of miRNA utilizing the droplet digital polymerase chain reaction quantification (ddPCR) [75,76,77]. The ddPCR is a method of nucleic acid amplification considered to be more sensitive than conventional real time quantitative PCR. Furthermore, it does not require any internal or external normalization, which improves its reliability [78,79].The results of those studies are presented in Table 1.

## 5. Prognostic Value of Circulating miRNA

We investigated 21 studies on the prognostic value of miRNA [28,32,35,36,37,46,48,52,53,61,64,65,66,72,74,80,81,82,83,84,85]. The prognostic value of different miRNAs was evaluated 30 times. In 24 cases, the prognostic value of the tested miRNA was confirmed with univariate analysis, which later was confirmed as an independent prognostic factor by multivariate Cox analysis in 17 cases. The most commonly tested circulating miRNA was miR-21, the up-regulated expression of which was associated with poor patient prognoses (5 studies). The same was also reported for miR-196a (2 studies). The details are provided in Appendix A. In a study by Gablo et al. [82], a panel consisting of down-regulated miR-99a-5p, miR-365a-3p and up-regulated miR-200c-3p had been proven to identify patients with poor prognosis after surgical resection with the sensitivity of 85% and specificity of 80%. According to the study by Vietsch et al. [86], two up-regulated miRNAs (miR-99a-5p and miR-125b-5p) are significantly associated with a shorter progression free survival (PFS).

## 6. Predictive Value of Circulating miRNA

The predictive value of circulating miRNA was evaluated in six studies covered here [46,83,84,85,87,88]. In a study by Miyamae et al. [53], an increased expression of plasma miR-744 indicated the tendency to worse PFS during gemcitabine therapy (*p* = 0.0533). Wang et al. [85] presented similar results for serum miR-21, the over-expression of which indicated a shortened time to progression (TTP) and overall survival (OS) among patients with PDAC stage III and IV during gemcitabine therapy (*p* = 0.029 for OS, CS III; *p* = 0.069 for OS, CS IV; *p* = 0.008 for TTP, CS III; *p* = 0.029 for TTP, CS IV). In a study by van der Sijde et al. [84], a high level of serum miR-373-3p before the start of treatment and a low level of serum miR-194-5p after one cycle of treatment were associated with early tumor progression during FOLFIRINOX therapy. In a study Demiray et al. [88], patients with low let-7c expression progressed significantly earlier than patients with high let-7c expression during FOLFIRINOX therapy. In study by Lu et al. [87], plasma miR-20a-5p expression was significantly lower in gemcitabine-resistant patients with PDAC than in non-resistant patients (*p* < 0.01). Kong et al. provided results that serum levels of miR-196a might be a useful marker for differentiating respectable and unresectable PDAC [83].

## 7. Screening Value of Circulating miRNA

In studies by Duell et al. [89] and Franklin et al. [38], the screening utility of circulating miRNA was investigated. In the nested case-control study by Duell et al. [89], six plasma miRNAs expression levels (miR-10a, -10b, -21-5p, -30c, -155 and -212) were statistically significantly associated with the risk of PDAC. However, in Franklin et al.’s. study, 15 miRNA candidate, that were differently expressed during initial screen (miR-574-3p, -885-5p, -144-3p, -130b-3p, -34a-5p, -24-3p, -106b-5p, -22-5p, -451a, -101-3p, -26a-5p, -197-3p, -423-3p, -122-5p, and let-7d-3p) were not altered in plasma before the diagnosis of PDAC, which left the authors with a conclusion that circulating miRNA expression levels change late after the diagnosis of PDAC [38].

## 8. Monitoring Therapies with Circulating miRNA

The usefulness of circulating miRNAs in therapy monitoring was addressed in 8 studies included in this review. Details can be found in Table 2. Most of the studies (7) evaluated the expression of miRNAs pre- and postoperatively, proving a change in expression levels. In one study, the expression of preoperatively down-regulated miRNA increased after a successful surgery [43] while in six studies the expression of preoperatively up-regulated miRNAs decreased after surgery [28,38,41,46,50,90,91]. The study by Meijer et al. [90] provided data on miR-181a-5p usefulness in therapy monitoring during systemic therapy with the FOLFIRINOX regimen.

## 9. Challenges to the Analysis of Circulating miRNA

Circulating miRNAs are good candidates for biomarkers. The relatively low risk of complications and minimally invasive and painless sampling procedure are what makes them potentially useful in cancer diagnosis, prognosis and therapy monitoring. However, the quantification of circulating miRNA might be challenging.

Although circulating miRNAs are considered rather stable molecules [92], proper measures should be taken during specimen sample collection and handling. Contamination with background cellular fraction of miRNAs caused by blood cell lysis remains a cause of severe preclinical bias. Collected samples should be processed within two hours. Plasma and serum could be placed at 4 °C for up to 24 h or preferably frozen at −80 °C for long-term storage [93]. What is more, the variability of circulating miRNA expression could also be influenced by BMI, sex and age. Thus, a proper experiment protocol is recommended [94].

Specimen control for hemolysis should involve a visual inspection or the spectrophotometric test of hemoglobin absorbance at 414 nm [95], with an absorbance cut-off value of >0.2 for a hemolyzed specimen. Samples that do not meet the criteria should be excluded from further analysis.

The spike-in control, which is a foreign small RNA molecule added to the sample before extraction, can be used for extraction efficiency control. The molecule is the same length as native miRNAs, but it lacks the sequence homology to endogenous miRNAs. It usually comesfrom an unrelated species, such as *Caenorhabditis elegans.* The use of such control can identify material losses [96]. Positive endogenous miRNAs controls are also used to estimate the extraction procedure. The miRNAs suitable for endogenous control should be present at stable and high levels in tested specimen and should not be affected by cellular material. This kind of control might be used alongside exogenous spike-in controls [97].

Among traditional high-throughput methods of miRNA analysis, such as microarrays or RNA sequencing (RNA-Seq), RT-Q-PCR with its high specificity and sensitivity is considered a gold standard. However, the main limitation of this approach is that only the targeted miRNAs can be quantified. It also requires predesigned RT-primers, the high costs of which might be a limiting factor as well [98].

It should be recommended for authors who plan future studies on clinical utility of circulating miRNAs to select an appropriate control group, set an independent intrinsic validation and perform an analysis of background factors influencing the levels of detected circulating miRNAs, most importantly age and sex, as presented in Takinawa et al.’s study [99].

## 10. Discussion

PDAC is surgically resectable in fewer than 20% of patients with PDAC [100]. A late diagnosis leads to a more serious clinical stage of the disease and a low 5-year survival rate [3]. The prognosis of PDAC has not improved over the past 20 years, with relatively stable mortality. Most patients are diagnosed in clinical stage IV of the disease, which means they are suitable for palliative treatment, with the median overall survival (mOS) estimated at 8 to 11 months. Since the overall survival depends mainly on the disease’s clinical stage, a search for methods with an efficient diagnosis of early PDAC is becoming more important.

Various miRNAs play important roles in the carcinogenesis and development of various cancers. Since circulating miRNAs emerged as a novel biomarker in various neoplasms [101,102,103] and are stable in bodily fluids in patients with PDAC, this review aims at providing an updated perspective on circulating miRNAs’ role in pancreatic cancer diagnosis. The single biomarker, the diagnostic utility of which was most frequently evaluated, was miR-21. Its diagnostic utility was examined in 7 studies reviewed here. The other commonly evaluated biomarkers were miR-155, miR-196a and miR-210, each of them present in 5 studies (S2).

The area under the curve (AUC) in receiver operating characteristics analysis (ROC) is used for evaluating the accuracy of diagnostic test. A result above 0.9 indicate a high diagnostic value. The diagnostic utility of circulating miRNA has been already established by multiple studies and meta-analyses [104,105] with miR-21 as a reliable biomarker in digestive system neoplasms with its 72% sensitivity, 82% specificity and an overall AUC of 0.86. However, the meta-analysis by Aalami et al. [106] demonstrated that single circulating miR-21 levels are not specific for subgroups of gastrointestinal cancers, such as esophageal squamous cell cancer, gastric cancer, pancreatic ductal adenocarcinoma or colorectal cancer. This meta-analysis’s limitations were high heterogeneity and publication bias. Most of the studies included in this review provided data of circulating miRNAs surpassing the commonly used CA19-9 as a biomarker in PDAC. However, some presented a poorer effect. To overcome the limitations of the diagnostic performance of single circulating miRNAs, they were combined into panels with CA19-9, other miRNAs or other useful biomarkers (S3 and S4)in several studies. The diagnostic panels reached high diagnostic value(AUC values above 0.9) in more cases than single miRNAs. In their meta-analysis, Pei et al. proved that multiple miRNAs combined into diagnostic panels provide a more accurate method of PDAC diagnosis than single circulating miRNAs. The diagnostic accuracy of panels of multiple miRNAs was verified in 19 studies. The sensitivity of such panels was 85% while the specificity was 89%. The AUC was determined to be 0.93. Compared to the diagnostic accuracy of a single miRNA in the 17 studies(sensitivity 78%, specificity 79% and AUC 0.84), combined miRNA panels provide a more accurate method of PDAC diagnosis. However, this meta-analysis had its limitations—most importantly, high heterogeneity among the reviewed studies caused by differences in miRNA profiling methods and specimen resources [107]. Moreover, the same miRNAs are deregulated in different types of neoplasm, meaning that they are not specific to any cancer type and are involved in various oncogenic pathways. For example, miRNA-21 is also involved in the carcinogenesis of colorectal cancer, breast cancer and glioblastomas [108]. Taking it into account, a combination of miRNAs can increase the accuracy of diagnostic panel regarding a specific cancer type. However, creating a complex diagnostic panel that could be implemented for clinical usage might be a costly and time-consuming procedure, which could be detrimental to its clinical utility.

Several prognostic factors are commonly used in pancreatic cancer therapy. Factors like histopathological type (adenosquamous carcinoma and undifferentiated carcinoma are considered pancreatic cancer types with a poorer prognosis than PDAC), high expression of CA 19-9 (in terms of survival after surgery, CA19-9 levels above 500 IU are related to a poor prognosis) or loss of SMAD 4 (mothers against decapentaplegic homolog 4) expression (poor prognosis) are implemented into treatment guidelines [109]. The search for other prognostic factors is important for the therapeutic process to determine which patients might benefit from specific therapies. For example, inflammation-related factors, such as neutrophil-to-lymphocyte ratio (NLR) play an increasing role in PDAC therapy since they are easy to obtain and relatively not expensive [110]. The prognostic value of circulating miRNA in PDAC is established in most studies as well. Similarly to circulating miRNAs’ diagnostic role, the miR-21 was the most frequently evaluated biomarker in terms of its prognostic performance. Three out of six studies reported miR-21 as an independent prognostic factor in multivariable regression analysis with *p* < 0.01.A similar effect was also detected for miR-196a (two studies, one of which included a multivariate analysis). In one study, the level of miR-21 was not associated with PDAC patients prognosis and, what is noteworthy, none of the six other tested miRNA expression levels was associated with PDAC prognosis [69] (S5). The authors suggest a longer time of observation and larger groups of enrolled patients could increase the value of further studies on this matter. The Guraya et al.’s meta-analysis infers that miR-21 constitutes a useful prognostic biomarker in digestive system neoplasms. The pooled hazard ratio (HR) of worse OS in patients with PDAC was 3.77 (1.63–8.73, *p* < 0.01). However, the meta-analysis has its limitations, with a high heterogeneity of the covered studies as the critical one [111].

The most intriguing role of miRNAs would be in screening patients with an increased risk of PDAC. Some of the markers in this review showed the potential for PDAC screening [89]. However, one study presented different results [38]. The number of studies evaluating this aspect of circulating miRNAs included in this review is low. It is probably due to the difficult methodology of such studies and a long time of patient follow-up. The differences between those studies are important. The study by Duell et al. was a prospective cohort study while in Franklin’s study, blood samples were retrieved from a biobank from retrospectively analyzed patients who had undergone pancreatic surgery. It seems that Duell’s study remains the only one evaluating the risk of pancreatic cancer among patients with an increased expression inmiRNA. With its prospective design and specimen collection many years before the diagnosis, it provides strong evidence for miRNAs screening value. However, the main qualities of the screening test should be its high specificity, safety and relatively low costs [112,113]. For now, circulating miRNAs testing seems to be unable to satisfy the cost criterion to be used as a population screening test. Therefore, the screening role of circulating miRNAs in PDAC remains uncertain and requires large, multicenter studies to provide reliable data.

The predictive value of circulating miRNAs indicates its utility to choose the most suitable therapy for the patient, either the surgical or systemic approach. The identification of patients that may respond to a certain therapy is crucial to improve treatment outcomes. This may also influence the decision to discontinue the therapy in patients who will no longer benefit from treatment. Studies included in this review provide data on this matter [46,83,84,85,87,88]. However, many factors contribute to the choice of therapy and the most significant of them is the patient’s performance status (PS) before the start of the treatment. Circulating miRNAs seem to be a promising factor. Still, further research is needed to determine their predictive role, especially in a prospective approach.

There are no specific biomarkers to monitor the response to treatment in PDAC. CA 19-9 expression has been suggested as a predictor for survival in patients with PDAC, but its decrease during systemic treatment is not associated with prolonged survival compared to patients who did not exhibit a significant decrease [114]. Therefore, the search for biomarkers, which may improve the decision-making process in PDAC treatment, is also relevant. Most of the studies covered in this analysis referred to circulating miRNAs changes in reference to surgical treatment. Expression of circulating miRNAs in postoperative specimen was significantly reduced compared to preoperative serum or plasma expression. These findings suggest that miRNAs are released by cancerous tissue and could be used to monitor tumor dynamics, such as the presence of a residual tumor or disease recurrence [28,38,41,46,50,91]. The study by Meijer et al. [90] demonstrated that miR-181a-5p is significantly down-regulated in patients with non-progressive PDAC compared to patients with progressive PDAC during FOLFIRINOX treatment. No such observation was made for patients treated with gemcitabine plus nab-paclitaxel. Notably, patients treated with the gemcitabine plus nab-paclitaxel regimen were older and had a worse PS, which could affect the result. Therefore, large-scale, prospective research is called for to establish circulating miRNA’s role in tumor response monitoring in PDAC.

Circulating miRNA constitutes a challenging biomarker to be implemented in clinical practice. First of all, its expression may be influenced by different factors like diet, medications, patient sex or the time of sample collection during the day. What is more, its stability relies on sample handling and storage. The quality control of extracted miRNA may be performed with endogenous miRNA controls or exogenous spike-in controls [115,116]. The studies handled in this review employed various quality control methods, the most common being RNU6B and cel-miR-39 (S2). Further research is required to determine the optimal protocol for circulating miRNA detection so that study results can be compared more easily.

## 11. Conclusions and Future Research

This review provides general information about circulating miRNAs diagnostic, prognostic and therapy monitoring value, which seems to be well-established in numerous studies. The screening value of circulating miRNA in PDAC remains uncertain and has to be further investigated. Due to the high mortality rate of pancreatic cancer patients, it is crucial to search for early diagnostic biomarkers to improve the survival rate.

There are many obstacles to employing circulating miRNA in clinical practice. A proper study design, reducing the risk of preclinical variability of miRNAs expression and standardized sample processing seem to be crucial for reducing the bias and may increase the reproducibility of the method. Standardization of diagnostic procedures for circulating miRNA expression should be the future aim. It would facilitate not only an easier juxtaposition of results, but also precise cancer diagnosis and prognosis. What is more, combining miRNAs into diagnostic panels seems to yield a more accurate method than using single miRNAs. Introducing other biomarkers, such as CA19-9, CEA, cytokines or proteins, into miRNAs diagnostic panels might improve their diagnostic accuracy. In terms of screening value, further research is needed to determine the panel with the highest specificity. Large-scale, prospective studies are required to determine predictive and therapy monitoring utility of circulating miRNAs.

After developing standardized and universally accepted methods for circulating miRNA testing in the future, these markers could be the novel additional diagnostic, prognostic and disease monitoring factor useful in clinical practice. However, further research is required to determine the optimal methods for circulating miRNA testing.

## Figures and Tables

**Figure 1 ijms-24-05113-f001:**
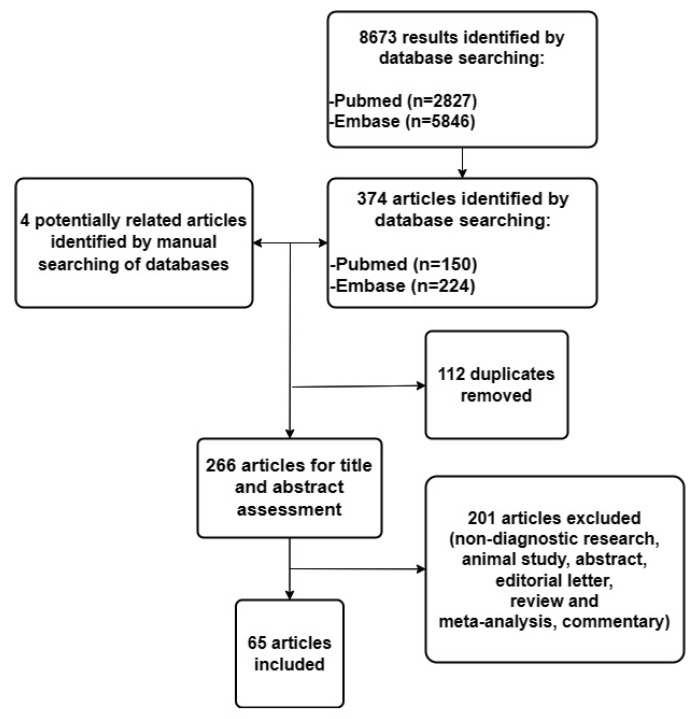
Flow diagram of the literature search and study selection process.

**Table 1 ijms-24-05113-t001:** Diagnostic performance of single miRNAs using the ddPCR method for expression evaluation, CS–clinical stage, AUC–area under the curve, SEN–sensitivity and SPE-specificity.

miRNA	Author	Source of Sample	Expression Level Compared to Non-Cancer Controls.	CS of PC Patients	Control	AUC	SEN	SPE	Diagnostic Performance Compared to CA19-9	AUC of Tested miRNA Combined with CA19-9	SEN with CA19-9	SPE with CA19-9
miR-1290	Tavano et al. [75]	plasma	Up- regulated	I-IV	Healthy subjects	0.734 [0.678–0.789]	56.3%	89.5%	Worse results	0.956 [0.933–0.979]	88.6%	96.6%.
miR-1273g-3p	Mazza et al. [76]	plasma	Up- regulated	I-IV	Healthy subjects	0.703 [0.639–0.768]	77.7%	57.3%	Worse results	0.940 [0.909–0.972]	90.2%	87.3%
miR-122-5p	Mazza et al. [76]	plasma	Up- regulated	I-IV	Healthy subjects	0.658 [0.590–0.726]	49.1%	79.2%	Worse results	0.933 (0.899–0.967)	83.9%	94.6%
let-7d	Suzuki et al. [77]	serum	Down- regulated	I-IV	Non-tumour controls (chronic pancreatitis, billiary stone, others)	0.83 [0.740–0.910]	888.9%	68.2%	Better results	-	-	-

**Table 2 ijms-24-05113-t002:** Therapy monitoring with circulating miRNA.

miRNA	Author	Source of Sample	Therapy Monitoring Utility	*p-*Value
miR-107	Imamura et al. [43]	plasma	up-regulation after surgery	*p* = 0.0186
miR-181a-5p	Meijer et al. [90]	plasma	down-regulation after treatment with FOLFIRINOX in non-progressive patients, compared with progressive patients	*p* = 0.003
miR-18a	Morimura et al. [91]	plasma	down-regulation after surgery	*p* = 0.0077
miR-196a	Slater et al. [57]	serum	down-regulation after surgery (9 cases)	-
miR-196b	Slater et al. [57]	serum	down-regulation after surgery (9 cases)	-
miR-221	Kawaguchi et al. [46]	plasma	down-regulation after surgery	*p* = 0.018
miR-223	Komatsu et al. [53]	plasma	down-regulation after surgery	*p* = 0.0297
miR-483-3p	Abue et al. [28]	plasma	down-regulation after surgery (2 cases)	-
miR-744	Miyamae et al. [53]	plasma	down-regulation after surgery	*p* = 0.0063

## Data Availability

The data used to support the findings of this study are available from the corresponding author upon request.

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
