# Peer review of "Clinical Value of Circulating miRNA in Diagnosis, Prognosis, Screening and Monitoring Therapy of Pancreatic Ductal Adenocarcinoma–A Review of the Literature"

_ijms, 2023, doi:10.3390/ijms24065113_

Round 1

Reviewer 1 Report

The review by Wnuk et al entitled "" is a very comprehensive review of a very important aspect of PC biology and will contribute to helping our understanding of the disease and also for clinicins in their work.

It is well presented and for the most comprenhenible, however the English needs to be looked over and corrected.

Just two examples

Abstract: they wrote: "Pancreatic cancer (PC) is considered 7th most...". They must write either "Pancreatic cancer (PC) is considered to be the 7th most..." or "Pancreatic cancer (PC) is the 7th most..."

Intro: they wrote: "PC was responsible for 466 003 deaths in 2020, what puts it as 7th leading..." substitute "what" with 'which'

The problem throughout the manuscript is at this level which (not what) is not so bad and easily corrected.

After the authors have done this I consider this excellent work ready to be published

Author Response

RESPONSE TO REVIEWERS

Manuscript: Clinical value of circulating miRNA in diagnosis, prognosis, screening and monitoring therapy of pancreatic ductal adeno-carcinoma – a review of literature.

Authors: Jakub Wnuk, Joanna Katarzyna Strzelczyk and Iwona Gisterek

We would like to thank the Reviewers for their valuable and detailed comments, suggestions and their time spent on reviewing the manuscript. We believe that after completion of the suggested edits, the revised manuscript has improved in the overall presentation and clarity.

Looking forward hearing from you soon.

Sincerely,

Jakub Wnuk

The review by Wnuk et al. entitled "" is a very comprehensive review of a very important aspect of PC biology and will contribute to helping our understanding of the disease and also for clinicins in their work.

It is well presented and for the most comprenhenible, however the English needs to be looked over and corrected.

Just two examples

Abstract: they wrote: "Pancreatic cancer (PC) is considered 7th most...". They must write either "Pancreatic cancer (PC) is considered to be the 7th most..." or "Pancreatic cancer (PC) is the 7th most..."

Intro: they wrote: "PC was responsible for 466 003 deaths in 2020, what puts it as 7th leading..." substitute "what" with 'which'

The problem throughout the manuscript is at this level which (not what) is not so bad and easily corrected.

After the authors have done this I consider this excellent work ready to be published.

We , have changed the examples provided by the Reviewer. We have also had the manuscript corrected by language editing service. If the manuscript needs more editing services, we will use some, provided by MDPI.

Reviewer 2 Report

This review introduces the detection of circulating miRNAs for screening, diagnosis, prognosis and monitoring therapy of pancreatic ductal adenocarcinoma. The characterization, application, future perspectives and challenges of circulating miRNAs-based pancreatic ductal adenocarcinoma evaluation are discussed in the manuscript. Considering the quality of the manuscript, I would not recommend the publication of this paper on IJMS unless a major revision has been made to address the issues of the article. Here are my comments:

1.       The manuscript mentioned that “Early diagnosis of PC is crucial for improving treatment outcomes. Ultrasound (US), computed tomography (CT), magnetic resonance imaging (MRI), endoscopic ultrasonography (EUS) with EUS-guided fine-needle aspiration (EUS-FNA) are conventional methods used in PC diagnosis” Then what is the characterization and limit of each method? I would recommend providing a table summarizing the advantages and disadvantages of these methods and comparing them with circulating miRNAs-based detection.

2.       As described in the manuscript “What is more, the variability of circulating miRNA expression could be also influenced by BMI, gender and age. Thus, the proper experiment protocol is recommended [94]. ” However, the proper experiment protocol cannot be found in the manuscript. I would recommend to show this protocol in a diagram or figure to directly reveal the standard procedure.

3.       More details are expected for some examples mentioned in the manuscript. For instance, it is stated that “In their meta-analysis, Pei et al. have proved that multiple miRNAs combined into diagnostic panels provide more accurate method of PDAC diagnosis [105].” How accurate is the meta-analysis? How many times of accuracy improvement when compared with conventional methods? These detailed information is expected.

4.       It is described that “These findings suggest that miRNAs are released by cancerous tissue and could be used to monitor tumor dynamics, such as presence of residual tumor or disease recurrence” I would recommend providing data proving the relationship between released miRNA and survival rate of PC patients.

Author Response

RESPONSE TO REVIEWERS

Manuscript: Clinical value of circulating miRNA in diagnosis, prognosis, screening and monitoring therapy of pancreatic ductal adeno-carcinoma – a review of literature.

Authors: Jakub Wnuk, Joanna Katarzyna Strzelczyk and Iwona Gisterek

We would like to thank the Reviewers for their valuable and detailed comments, suggestions and their time spent on reviewing the manuscript. We believe that after completion of the suggested edits, the revised manuscript has improved in the overall presentation and clarity.

Looking forward hearing from you soon.

Sincerely,

Jakub Wnuk

This review introduces the detection of circulating miRNAs for screening, diagnosis, prognosis and monitoring therapy of pancreatic ductal adenocarcinoma. The characterization, application, future perspectives and challenges of circulating miRNAs-based pancreatic ductal adenocarcinoma evaluation are discussed in the manuscript. Considering the quality of the manuscript, I would not recommend the publication of this paper on IJMS unless a major revision has been made to address the issues of the article. Here are my comments:

  1. The manuscript mentioned that “Early diagnosis of PC is crucial for improving treatment outcomes. Ultrasound (US), computed tomography (CT), magnetic resonance imaging (MRI), endoscopic ultrasonography (EUS) with EUS-guided fine-needle aspiration (EUS-FNA) are conventional methods used in PC diagnosis” Then what is the characterization and limit of each method? I would recommend providing a table summarizing the advantages and disadvantages of these methods and comparing them with circulating miRNAs-based detection.

We have added the table into the manuscript providing information about advantages and disadvantages of the mentioned methods. However, the circulating miRNAs-based detection methods were not compared to ultrasound, computed tomography, magnetic resonance imaging, endoscopic ultrasonography with EUS-guided fine-needle aspiration in any of cited studies, thus we can not compare them in our review. Unfortunately, our review could provide only general information on diagnostic performance of circulating miRNAs in pancreatic ductal adenocarcinoma.

To facilitate the work of the Reviewers, we attached a marked-up version of the manuscript.

Supplementary Table S1. Characteristics of classic diagnostic methods of pancreatic ductal adenocarcinoma.

Diagnostic tool

Advantages

Disadvantages

Transabdominal ultrasound

·        non-invasive

·        relatively cost-effective

·        difficulties in imaging of pancreatic body and tail cancers

·        highly dependent on the operator’s experience

Computed tomography

·        non-invasive

·        good spatial resolution

·        good temporal resolution

·        wide anatomic coverage

·        best performance for the evaluation of vascular involvement

·        may not depict small metastases to the liver or peritoneum

EUS-guided fine-needle aspiration

·        high-resolution images of the pancreas

·        ability to obtain specimens for histopathological diagnosis

·        an invasive procedure, however considered safe and accurate

Magnetic resonance imaging

·        non-invasive

·        greater soft-tissue contrast of MRI compared with that of CT

·        worse spatial resolution compared to CT (requires relatively more time to perform the imaging)

  1. As described in the manuscript “What is more, the variability of circulating miRNA expression could be also influenced by BMI, gender and age. Thus, the proper experiment protocol is recommended [94]. ” However, the proper experiment protocol cannot be found in the manuscript. I would recommend to show this protocol in a diagram or figure to directly reveal the standard procedure.

Circulating miRNAs expression studies are developing fast. However, several studies have provided data that several factors such as blood sample collection and processing methods, age, sex, and other physiological conditions, can affect the normal levels of circulating miRNAs. Some studies try to summarize and address those problems, unfortunately they do not provide one solution with single standard procedure.

A study by Takizawa et al. entitled “Circulating microRNAs: Challenges with their use as liquid biopsy biomarkers.” (Cancer Biomark. 2022;35(1):1-9. doi: 10.3233/CBM-210223. PMID: 35786647; PMCID: PMC9661319), try to present this problem. Therefore we add the following part into the manuscript, following your advice.

It should be recommended for authors, who plan future studies on circulating miRNAs clinical utility to select an appropriate control group, set an independent in-trinsic validation and perform an analysis of background factors influencing levels of detected circulating miRNAs, most importantly age and sex, as presented in Takinawa’s et al. study [98]

  1. More details are expected for some examples mentioned in the manuscript. For instance, it is stated that “In their meta-analysis, Pei et al. have proved that multiple miRNAs combined into diagnostic panels provide more accurate method of PDAC diagnosis [105].” How accurate is the meta-analysis? How many times of accuracy improvement when compared with conventional methods? These detailed information is expected.

It is an important issue, which Reviewers have pointed out. The problems with establishing single protocol with one intrinsic control and background factors influencing the results are also visible in meta-analyses. Most of them report high heterogeneity and publication biases. Unfortunately, studies included into this study are influenced by those biases as well. Those studies do not compare the accuracy of circulating miRNA to conventional methods.

We add and change the following parts of the manuscript, to provide more specific information:

The diagnostic utility of circulating miRNA has been already established by multiple studies and meta-analyses [103, 104] with miR-21 as a reliable biomarker in digestive system neoplasms with its 72% sensitivity, 82% specificity and the overall AUC of 0.86. However, the meta-analysis by Aalami et al. provided data that single circulating miR-21 levels is not specific for subgroups of gastrointestinal cancers such as esophageal squamous cell cancer, gastric cancer, pancreatid ductal anenocarcinoma or colorectal cancer. This meta-analysis had its limitations as well with high heterogenity and publication bias.

In their meta-analysis, Pei et al. have proved that multiple miRNAs combined into diagnostic panels provide more accurate method of PDAC diagnosis than single circulating miRNAs The diagnostic accuracy of panels of multiple  miRNAs was performed in in 19 studies. The sensitivity of such panels was 85% while the specificity was 89%. The AUC was determined to be 0.93 Compared to the diagnostic accuracy of a single miRNA in the 17 studies (sensitivity 78%, specificity 79% and the AUC 0.84), combined miRNA panel provide more accurate method of PDAC diagnosis. However this metaanalysis had its limitations, most importantly high heterogeneity among the included studies, caused by differences in miRNA profiling methods; specimen resources.

  1. It is described that “These findings suggest that miRNAs are released by cancerous tissue and could be used to monitor tumor dynamics, such as presence of residual tumor or disease recurrence” I would recommend providing data proving the relationship between released miRNA and survival rate of PC patients.

Thank you, for pointing out this issue. We should have mentioned this in the “prognostic value” part of our manuscript. Therefore we change this parts:

We include 21 studies into the analysis that referred to prognostic value of miRNA [28, 32, 64–66, 72, 74, 80–84, 35, 85, 36, 37, 46, 48, 52, 53, 61], which evaluated prognostic value of different miRNAs 30 times. In 24 cases, prognostic value of tested miRNA was confirmed by univariate analysis, which later was confirmed as independent prognostic factor by multivariate Cox analysis in 17 cases. Most commonly tested circulating miRNA was miR-21, which up-regulated expression was associated with poor prognosis of patients (5 studies). This was also reported for miR-196a (2 studies). The details are pro-vided in Supplementary Table 5(S5).

The prognostic value of circulating miRNA in PDAC is established in most studies as well. Similarly to circulating miRNAs diagnostic role, the miR-21 was the most frequently evaluated biomarker in terms of its prognostic performance. Three out of six studies reported miR-21 as an independent prognostic factor in multivariable regression analysis with p values <0.01 Similar effect was also detected for miR-196a (2 studies, one of them performed a multivariate analysis). In one study, level of miR-21 was not associated with PDAC patients prognosis and, what is noteworthy, none of 6 other tested miRNA expression levels was associated with PDAC prognosis [64] (S5).. The authors suggest that longer time of observation and larger groups of patients enrolled to the studies could increase the value of further studies on this matter. The Guraya S. et al. meta-analysis infers that miR-21 constitutes a useful prognostic biomarker in digestive system neoplasms. The pooled hazard ratio (HR) of worse OS in patients with PDAC was 3.77 (1.63–8.73, p value < 0.01). However, the meta-analysis  have its limitations, with high heterogeneity of included studies to be mentioned as the most important one. [110].

We will also have our manuscript corrected by language editing services, therefore the changes mentioned above could be changed in the final form of the draft.

Reviewer 3 Report

The review entitled 'Clinical value of circulating miRNA in diagnosis, prognosis, screening, and monitoring therapy of pancreatic ductal adenocarcinoma' discusses the increasing incidence of pancreatic cancer, the importance of early diagnosis, and the potential role of circulating microRNAs as diagnostic and prognostic biomarkers for pancreatic ductal adenocarcinoma. The article highlights the value of these biomarkers for screening, diagnosis, prognosis, and therapy monitoring of PDAC. It is a timely contribution to the filed, and the review aims to evaluate the evidence supporting the use of circulating miRNA as a clinical tool in various areas related to pancreatic ductal adenocarcinoma, which could ultimately contribute to improving patient outcomes. However, there are some major concerns and important points, which are missing and needs to be addressed. My comments are appended below:

Abstract

Pancreatic ductal adenocarcinoma (PDAC) is abbreviated twice in abstract, correct and used only once through the manuscript.

In introduction, what is the clinical significance of circulating miRNAs in the diagnosis of pancreatic ductal adenocarcinoma (PDAC), must be discussed as background information. Circulating miRNAs have gained attention as potential non-invasive diagnostic biomarkers for PDAC. They can be detected in body fluids, such as blood, serum, and plasma, and their expression profiles have been shown to differ between PDAC patients and healthy individuals. The clinical significance of circulating miRNAs in the diagnosis of PDAC lies in their ability to provide a minimally invasive alternative to the more invasive diagnostic methods currently available. These points must be considered as motivation for this review.

In table 1, control mention Healthy patients, does author mean patients enrolled other than PDAC? Clarify this.

Several studies have shown that certain circulating miRNAs are associated with poor prognosis in PDAC patients. For example, high levels of miR-21, miR-155, and miR-196a have been associated with worse overall survival. However, more research is needed to determine the overall effectiveness of circulating miRNAs as prognostic biomarkers for PDAC. Therefore, how effective are circulating miRNAs as prognostic biomarkers for PDAC? Discuss based on evidences collected n this study.

In Figure 1. Flow diagram of the literature search and study selection process, improve the font size to make the labels legible.

Circulating miRNAs have the potential to be used in screening for PDAC, particularly in high-risk populations, such as individuals with a family history of the disease. However, more research is needed to determine the sensitivity and specificity of circulating miRNAs in detecting early-stage PDAC. What is the role of circulating miRNAs in screening for PDAC, these specific and important points are missing in discussion?

Cite a latest report ´Among traditional high-throughput methods of miRNA analysis such as microarrays, RNA sequencing (RNA-Seq), RT-Q-PCR is considered a gold standard with its high specificity and sensitivity´ to make references up to date.

How do circulating miRNAs compare to other biomarkers in diagnosing PDAC? Circulating miRNAs have shown promise in diagnosing PDAC, but their diagnostic accuracy needs to be compared to that of other biomarkers, such as CA 19-9. In addition, more research is needed to determine the most effective miRNA panel for PDAC diagnosis.

Can circulating miRNAs help in predicting the response to therapy in PDAC patients since circulating miRNAs have the potential to be used as predictive biomarkers for therapy response in PDAC patients. For example, some studies have shown that miR-21 and miR-210 are associated with resistance to chemotherapy. However, more research is needed to determine the overall effectiveness of circulating miRNAs in predicting therapy response?

There are several challenges in utilizing circulating miRNAs as clinical biomarkers for PDAC. These include variability in miRNA expression levels between patients, the need for standardized methods for miRNA detection and analysis, and the need to validate miRNA panels in large patient cohorts. What are the challenges in utilizing circulating miRNAs as clinical biomarkers for PDAC needs to be included in the draft.

Sentence ´The area under the curve (AUC) in receiver operating characteristics analysis (ROC) is used for evaluating the accuracy of diagnostic test with result above 0.9 indicating high diagnostic value. ´needs a suitable reference regarding AUC and RIC, I encourage authors to cite a report https://doi.org/10.1186/1471-2350-13-70 on the topic along line 236-238 along with statement.

What is the current state of research on the use of circulating miRNAs in monitoring therapy for PDAC? There is ongoing research on the use of circulating miRNAs in monitoring therapy for PDAC, particularly in predicting therapy response and detecting recurrence. However, more research is needed to determine the most effective miRNA panels for these purposes.

In conclusion, add what are the potential future directions for research on circulating miRNAs and PDAC, which could include the development of more accurate and specific miRNA panels for diagnosis, prognosis, and therapy monitoring, as well as the investigation of miRNA-mediated mechanisms in PDAC tumorigenesis.

Provide a list of all abbreviations used on the manuscript as many of them used without full form, which makes it difficult to understand.

In order for the results of studies on circulating miRNAs to be translated into clinical practice, standardized methods for miRNA detection and analysis need to be developed and validated. How can the results of studies on circulating miRNAs be translated into clinical practice, add to conclusion?

Author Response

RESPONSE TO REVIEWERS

Manuscript: Clinical value of circulating miRNA in diagnosis, prognosis, screening and monitoring therapy of pancreatic ductal adeno-carcinoma – a review of literature.

Authors: Jakub Wnuk, Joanna Katarzyna Strzelczyk and Iwona Gisterek

We would like to thank the Reviewers for their valuable and detailed comments, suggestions and their time spent on reviewing the manuscript. We believe that after completion of the suggested edits, the revised manuscript has improved in the overall presentation and clarity.

Looking forward hearing from you soon.

Sincerely,

Jakub Wnuk

The review entitled 'Clinical value of circulating miRNA in diagnosis, prognosis, screening, and monitoring therapy of pancreatic ductal adenocarcinoma' discusses the increasing incidence of pancreatic cancer, the importance of early diagnosis, and the potential role of circulating microRNAs as diagnostic and prognostic biomarkers for pancreatic ductal adenocarcinoma. The article highlights the value of these biomarkers for screening, diagnosis, prognosis, and therapy monitoring of PDAC. It is a timely contribution to the filed, and the review aims to evaluate the evidence supporting the use of circulating miRNA as a clinical tool in various areas related to pancreatic ductal adenocarcinoma, which could ultimately contribute to improving patient outcomes. However, there are some major concerns and important points, which are missing and needs to be addressed. My comments are appended below:

Abstract

Pancreatic ductal adenocarcinoma (PDAC) is abbreviated twice in abstract, correct and used only once through the manuscript.

We have changed this part of abstract.

In introduction, what is the clinical significance of circulating miRNAs in the diagnosis of pancreatic ductal adenocarcinoma (PDAC), must be discussed as background information. Circulating miRNAs have gained attention as potential non-invasive diagnostic biomarkers for PDAC. They can be detected in body fluids, such as blood, serum, and plasma, and their expression profiles have been shown to differ between PDAC patients and healthy individuals. The clinical significance of circulating miRNAs in the diagnosis of PDAC lies in their ability to provide a minimally invasive alternative to the more invasive diagnostic methods currently available. These points must be considered as motivation for this review.

We have added this part to the manuscript:

Therefore, circulating miRNAs have gained more attention in recent years, as minimally invasive cancer markers, alternative and supplementary method for more invasive diagnostic methods currently available.

In table 1, control mention Healthy patients, does author mean patients enrolled other than PDAC? Clarify this.

Yes, this meant patients other than PDAC. We have changed this part of table.

Several studies have shown that certain circulating miRNAs are associated with poor prognosis in PDAC patients. For example, high levels of miR-21, miR-155, and miR-196a have been associated with worse overall survival. However, more research is needed to determine the overall effectiveness of circulating miRNAs as prognostic biomarkers for PDAC. Therefore, how effective are circulating miRNAs as prognostic biomarkers for PDAC? Discuss based on evidences collected n this study.

We have changed and added this part into the manuscript:

The prognostic value of circulating miRNA in PDAC is established in most studies as well. Similarly to circulating miRNAs diagnostic role, the miR-21 was the most frequently evaluated biomarker in terms of its prognostic performance. Three out of six studies reported miR-21 as an independent prognostic factor in multivariable regression analysis with p values <0.01 Similar effect was also detected for miR-196a (2 studies, one of them performed a multivariate analysis). In one study, level of miR-21 was not associated with PDAC patients prognosis and, what is noteworthy, none of 6 other tested miRNA expression levels was not associated with PRAC prognosis [64] (S54).. The authors suggest that longer time of observation and larger groups of patients enrolled to the studies could increase the value of further studies on this matter. The Guraya S. et al. meta-analysis infers that miR-21 constitutes a useful prognostic biomarker in digestive system neoplasms. The pooled hazard ratio (HR) of worse OS in patients with PDAC was 3.77 (1.63–8.73, p value < 0.01). However, the meta-analysis  have its limitations, with high heterogeneity of included studies to be mentioned as the most important one. [110].

In Figure 1. Flow diagram of the literature search and study selection process, improve the font size to make the labels legible.

We agree with this comment. The font size should be increased. We have changed this part of manuscript as well.

Circulating miRNAs have the potential to be used in screening for PDAC, particularly in high-risk populations, such as individuals with a family history of the disease. However, more research is needed to determine the sensitivity and specificity of circulating miRNAs in detecting early-stage PDAC. What is the role of circulating miRNAs in screening for PDAC, these specific and important points are missing in discussion?

We agree with this comment. We have tried to address this issue in this review, however we have managed to retrieve two studies addressing this matter, a study by Duell et al.[89] and Franklin et al.[30]. Their results remain contradicting. We have added the summary of this matter, by adding this part into the manuscript.

For now, circulating miRNAs testing seems to be unable to fulfill the cost criteria, to use it as  population screening test. The screening role of circulating miRNAs in PDAC remains uncertain, and requires large, multicenter studies, to provide reliable data.

Cite a latest report DOI: 10.3389/fgene.2022.1028081 ´Among traditional high-throughput methods of miRNA analysis such as microarrays, RNA sequencing (RNA-Seq), RT-Q-PCR is considered a gold standard with its high specificity and sensitivity´ to make references up to date.

We have added this report into our review. May we ask, if it is the correct one?
“D. Gemmati et al., “Host genetics impact on SARS-CoV-2 vaccine-induced immunoglobulin levels and dynamics: The role of TP53, ABO, APOE, ACE2, HLA-A, and CRP genes,” Front. Genet., vol. 13, p. 3145, Nov. 2022”

How do circulating miRNAs compare to other biomarkers in diagnosing PDAC? Circulating miRNAs have shown promise in diagnosing PDAC, but their diagnostic accuracy needs to be compared to that of other biomarkers, such as CA 19-9. In addition, more research is needed to determine the most effective miRNA panel for PDAC diagnosis.

Circulating miRNAs are usually compared to most commonly used biomarker in PDAC, which is CA-19.9 We included 16 studies in the analysis that involved analysis of diagnostic performance of single miRNA compared to diagnostic utility of CA19-9, with 55 single circulating miRNAs evaluated 59 times. In those studies 66.10% of them presented better diagnostic value than CA19-9 alone, comparing their AUC in ROC analyses.

Can circulating miRNAs help in predicting the response to therapy in PDAC patients since circulating miRNAs have the potential to be used as predictive biomarkers for therapy response in PDAC patients. For example, some studies have shown that miR-21 and miR-210 are associated with resistance to chemotherapy. However, more research is needed to determine the overall effectiveness of circulating miRNAs in predicting therapy response?

We believe that more research is needed to determine the predictive role of circulating miRNA in PDAC. Predictive value of circulating miRNA was evaluated in 6 included in ous analysis. For example Wang et al. provided data that overexpression of serum miR‐21 indicated shortened time-to progression (TTP) and overall survival (OS) among patients with PDAC stage III and IV during gemcitabine systemic therapy. 

There are several challenges in utilizing circulating miRNAs as clinical biomarkers for PDAC. These include variability in miRNA expression levels between patients, the need for standardized methods for miRNA detection and analysis, and the need to validate miRNA panels in large patient cohorts. What are the challenges in utilizing circulating miRNAs as clinical biomarkers for PDAC needs to be included in the draft.

We have added parts into the manuscript:

It should be recommended for authors, who plan future studies on circulating miRNAs clinical utility to select an appropriate control group, set an independent in-trinsic validation and perform an analysis of background factors influencing levels of detected circulating miRNAs, most importantly age and sex, as presented in Takinawa’s et al. study [98].

Further studies are required to determine an optimal protocol, for the circulating miRNA detection, to provide an easier way to compare study results.

Sentence ´The area under the curve (AUC) in receiver operating characteristics analysis (ROC) is used for evaluating the accuracy of diagnostic test with result above 0.9 indicating high diagnostic value. ´needs a suitable reference regarding AUC and RIC, I encourage authors to cite a report https://doi.org/10.1186/1471-2350-13-70 on the topic along line 236-238 along with statement.

May we ask if we have found a correct article to cite?
“Gemmati, D., Zeri, G., Orioli, E. et al. Polymorphisms in the genes coding for iron binding and transporting proteins are associated with disability, severity, and early progression in multiple sclerosis. BMC Med Genet 13, 70 (2012). https://doi.org/10.1186/1471-2350-13-70”

What is the current state of research on the use of circulating miRNAs in monitoring therapy for PDAC? There is ongoing research on the use of circulating miRNAs in monitoring therapy for PDAC, particularly in predicting therapy response and detecting recurrence. However, more research is needed to determine the most effective miRNA panels for these purposes.

In terms of systemic therapy only one study in this review evaluated this topic. This was the  study by Meijer et al. [90], which provided data on miR-181a-5p usefulness in therapy monitoring during systemic therapy with FOLFIRINOX regimen. It showed that mentioned miRNA, was significantly down-regulated after treatment with FOLFIRINOX in non-progressive patients, compared to progressive patients.

In conclusion, add what are the potential future directions for research on circulating miRNAs and PDAC, which could include the development of more accurate and specific miRNA panels for diagnosis, prognosis, and therapy monitoring, as well as the investigation of miRNA-mediated mechanisms in PDAC tumorigenesis.

Provide a list of all abbreviations used on the manuscript as many of them used without full form, which makes it difficult to understand.

We agree with this proposition. We have added the abbreviation list to the manuscript.

In order for the results of studies on circulating miRNAs to be translated into clinical practice, standardized methods for miRNA detection and analysis need to be developed and validated. How can the results of studies on circulating miRNAs be translated into clinical practice, add to conclusion?

For now, we find that circulating miRNAs could be an additional diagnostic and prognostic marker for PDAC. We add the following information to the conclusion:

After developing standardized and universally accepted methods of circulating miRNA testing, in the future, these markers could be the novel additional diagnostic, prognostic and disease monitoring factor, useful in clinical practice. However, further studies are required to determine the most optimal methods of circulating miRNA testing.

We will also have our manuscript corrected by language editing services, therefore the changes mentioned above could be changed in the final form of the draft.

Reviewer 4 Report

Comments:

1. Add each miRNA's function and target genes on the Table.

2. Any splicing factors associated with those miRNAs on the manuscript?

3. Any miRNA found in saliva as marker?

Author Response

RESPONSE TO REVIEWERS

Manuscript: Clinical value of circulating miRNA in diagnosis, prognosis, screening and monitoring therapy of pancreatic ductal adeno-carcinoma – a review of literature.

Authors: Jakub Wnuk, Joanna Katarzyna Strzelczyk and Iwona Gisterek

We would like to thank the Reviewers for their valuable and detailed comments, suggestions and their time spent on reviewing the manuscript. We believe that after completion of the suggested edits, the revised manuscript has improved in the overall presentation and clarity.

Looking forward hearing from you soon.

Sincerely,

Jakub Wnuk

Comments:

  1. Add each miRNA's function and target genes on the Table.

We have added miRNAs targets on Supplementary Table. However, in most cases, predicted miRNA target genes are numerous, therefore we placed there links to related databases with more precise information.

  1. Any splicing factors associated with those miRNAs on the manuscript?

Unfortunately, we did not covered this matter in our review.

  1. Any miRNA found in saliva as marker?

Unfortunately, we aimed to cover only the role of circulating miRNA in our review. However, salivary miRNAs, such as miR-3679-5p, miR-940, miR-21, miR-23a, miR-23b, miR-29 could be used as markers in PDAC as well and seem to be even less invasive than circulating miRNAs, detected in serum or plasma.

We will also have our manuscript corrected by language editing services, therefore the changes mentioned above could be changed in the final form of the draft.

Round 2

Reviewer 2 Report

The manuscript is well revised and I would recommend publication of the manuscript. Thank you for offering me the opportunity to review these works.

Reviewer 3 Report

accept

Reviewer 4 Report

No more comments